# MultiChartQA-R: A Benchmark for Multi-Chart Question Answering in Real-World Reasoning Scenarios

## Abstract

Existing benchmarks for chart analysis primarily focus on single-chart tasks, whereas multi-chart benchmarks are scarce and limited to simplistic question types, making it difficult to comprehensively evaluate the reasoning and decision-making capabilities of multimodal large language models (MLLMs) in realistic scenarios. We present **MultiChartQA-R**, a benchmark designed to evaluate multi-chart question answering capabilities, ranging from fundamental abilities to decision-making applications, with four progressively complex reasoning tasks that encompass real-world scenarios: cross-chart trend comparison, complementary data integration, anomaly and causal analysis, and strategy recommendation. The benchmark consists of versions in three major languages, each containing 695 chart–code pairs and 2,160 QA pairs, with extensibility to additional languages. We further propose a flexible multiple-choice evaluation metric that can be adjusted based on real-world scenarios, along with an extended dataset consisting of 512 charts and 1,212 QA pairs, designed to study retrieval and scaling behavior as the number of charts increases. Our evaluation of 13 representative MLLMs (4 proprietary models and 9 open-weight models) reveals significant performance gaps compared to human, especially in cross-chart visual perception, data integration, and aligning with human preferences. Additionally, our experiments reveal interesting multilingual characteristics of multi-chart question answering.

## 1 Introduction

Multimodal large language models (MLLMs) have recently demonstrated outstanding performance in various vision-language tasks, such as visual question answering (VQA) (Schwenk et al., 2022; Li et al., 2024b; Jia et al., 2025), chart-to-code generation (Yang et al., 2024), image captioning (Agrawal et al., 2019; Rahman et al., 2023; Kantharaj et al., 2022), and chart question answering (Masry et al., 2022; Methani et al., 2020; Wang et al., 2024; Li et al., 2025; Zeng et al., 2025). The task of chart question answering can be found everywhere in our daily work. Charts, as a powerful tool for data visualization, enable users to quickly grasp trends, patterns, and relationships within the data, thereby facilitating the formulation of strategies for subsequent actions. Many critical scenarios involve the comprehensive analysis of multiple charts. For example, in the financial sector, analysts examine several stock-related indicator charts to predict market trends; researchers compare multiple experimental data charts to discover patterns; and business managers analyze multiple charts related to departmental performance and costs to devise response strategies. However, the effectiveness of multimodal large language models in handling real-world multi-chart analysis scenarios remains insufficiently investigated.

Current chart analysis benchmarks mostly focus on single-chart tasks (Kahou et al., 2018; Kafle et al., 2018; Methani et al., 2020; Masry et al., 2022; Xu et al., 2023; Wang et al., 2024), primarily studying data extraction and multi-hop reasoning within a single chart. These benchmarks do not cover the multi-chart analysis scenarios encountered in real-world applications. The number of multi-chart benchmarks (Liu et al., 2024; Zhu et al., 2025b) is limited, and the variety of questions is insufficient. Research in this area primarily focuses on data comparison between charts and multi-hop question answering, with less emphasis on more complex cross-chart deep logical reasoning and multi-dimensional information integration. Moreover, existing benchmarks for multi-chart

analysis are predominantly English-centric, failing to meet the practical demands of multilingual chart analysis in a globalized context.

To address this, we introduce **MultiChartQAR** (fig. 1), a benchmark designed to evaluate multi-chart question answering abilities, from fundamental skills to decision-making applications. The core of multi-chart joint question answering lies in addressing questions that cannot be answered by a single chart alone, requiring the extraction, correlation, and reasoning of information across multiple charts. MultiChartQA-R is designed to reflect the practical scenarios of multi-chart question answering. It defines four tasks (section 2.1) to evaluate the capabilities of MLLMs. 1) **Cross-chart trend inference:** Emphasizes the ability of "information correlation," requiring the identification of dynamic relationships (e.g., synchronization, divergence) between indicators across different charts, which is fundamental for multi-chart analysis. 2) **Complementary data integration:** Focuses on the "data utilization" ability, emphasizing the extraction of hidden insights through logical or mathematical combinations of multi-chart data, highlighting the core value of "complementarity" in multi-chart data. 3) **Anomaly and pattern analysis:** Centers on the "deep analysis" ability, requiring the exploration of the underlying causes behind surface-level phenomena by combining chart information with external knowledge, thus reflecting the "depth" of the analysis. 4) **Strategy recommendation:** Focuses on the "practical application" ability, providing actionable decision-making suggestions based on the correlation patterns between multiple charts, thereby reflecting the "practicality" of the analysis. Together, these four tasks form a comprehensive logical chain for multi-chart analysis, progressing from "basic correlation" and "data utilization" to "deep analysis" and "practical application," covering the essential capabilities required for multi-chart joint question answering.

This paper presents the construction and expansion process of MultiChartQA-R (section 2.2), which includes the creation of chart-code pairs, QA pair generation, and multilingual expansion methods. MultiChartQA-R includes 14 chart types across 36 domains, available in three languages, with each language containing 2,160 QA pairs to ensure diverse coverage, with stringent data quality control and validation applied (section 2.3). A comprehensive comparison with existing chart QA benchmarks was performed (section 2.4), demonstrating MultiChartQA-R's significance in the field of chart-based question answering.

We evaluated four proprietary models and nine open-weight MLLMs on the MultiChartQA-R benchmark (section 3.1). To better evaluate model performance, we propose a flexible multiple-choice metric (appendix E.1) that balances rewards and penalties. This metric also allows for assessing the model's conservatism or aggressiveness, facilitating the training of models with different preferences. Extensive experiments show that proprietary and certain open-source models excel in analytical decision-making, but still lag behind humans in basic visual understanding and data integration (section 3.3). We conducted a comprehensive set of comparisons (section 4), exploring the performance of MLLMs in real-world multi-chart QA scenarios and examining the multilingual aspects of cross-chart question answering.

Our main contributions are as follows:

- We introduce the first scalable, multilingual benchmark for multi-chart question answering, designed to focus on real-world multi-chart task scenarios.

- We propose a flexible multiple-choice evaluation metric that balances rewards and penalties, reflecting the model's analytical decision-making ability and preferences. It can also be used for training preference-based models.

- We comprehensively evaluate existing MLLMs to provide insights into the critical capabilities required for real-world multi-chart scenarios and to assess their performance on cross-modal, multilingual multi-chart tasks.

## 2 MULTICHARTQA-R:

In this section, we first introduce the definition of four tasks involved in MultiChartQA-R (section 2.1), and then delineate the data curation process (section 2.2). Subsequently, we conduct a quantitative analysis to demonstrate the diversity of MultiChartQA-R and validate its quality through manual evaluation methods (section 2.3). Finally, we compare MultiChartQA-R with existing related benchmarks to demonstrate its superiority and effectiveness (section 2.4).

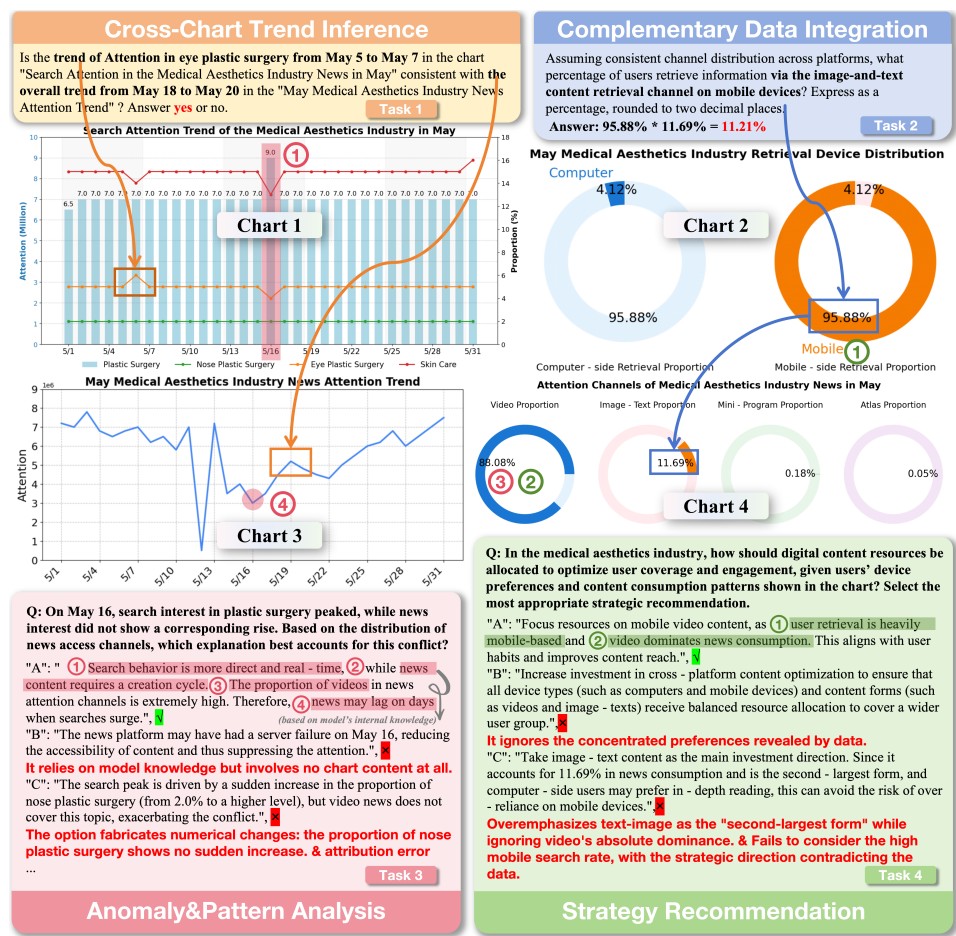

Figure 1: Examples of the four tasks in MultiChartQA-R. Task1&2 use arrows to illustrate the solution process, where the model must identify corresponding trends in the charts for comparison or find complementary data for integration. Task4&5 require identifying related information across multiple charts and reasoning based on the model's internal knowledge to derive the correct inference. Information used by the model is numbered, with erroneous inferences highlighted in red.

## 2.1 TASK DEFINITION

We designed four tasks based on common multi-chart question answering scenarios encountered in daily life and work. These tasks form a complete logical chain of multi-chart analysis, progressing from "basic correlation" and "data utilization" to "deep analysis" and finally "practical application," covering the core capability requirements of multi-chart question answering scenarios.

**Cross-Chart Trend Inference** Task 1 aims to evaluate the model's ability to analyze and judge trends across multiple charts, requiring the model to discern the relationships between the trends of various indicators. Specifically, the model needs to identify the trend directions (such as increasing, decreasing, or stable) of indicators in different charts and assess their synchronization or divergence, as shown in Task 1 of Figure 1.

**Complementary Data Integration** Task 2 evaluates the model's ability to integrate complementary data from different charts and derive hidden information through logical combinations or mathematical operations. These data may include proportions, totals, ratios, and other forms, and the desired result cannot be directly obtained from a single chart; instead, information from multiple charts must be combined for inference, as shown in Task 2 of Figure 1.

**Anomaly and Pattern Analysis** Task 3 requires the model to identify anomalous data phenomena or potential underlying patterns across multiple charts and provide explanations for these anomalies

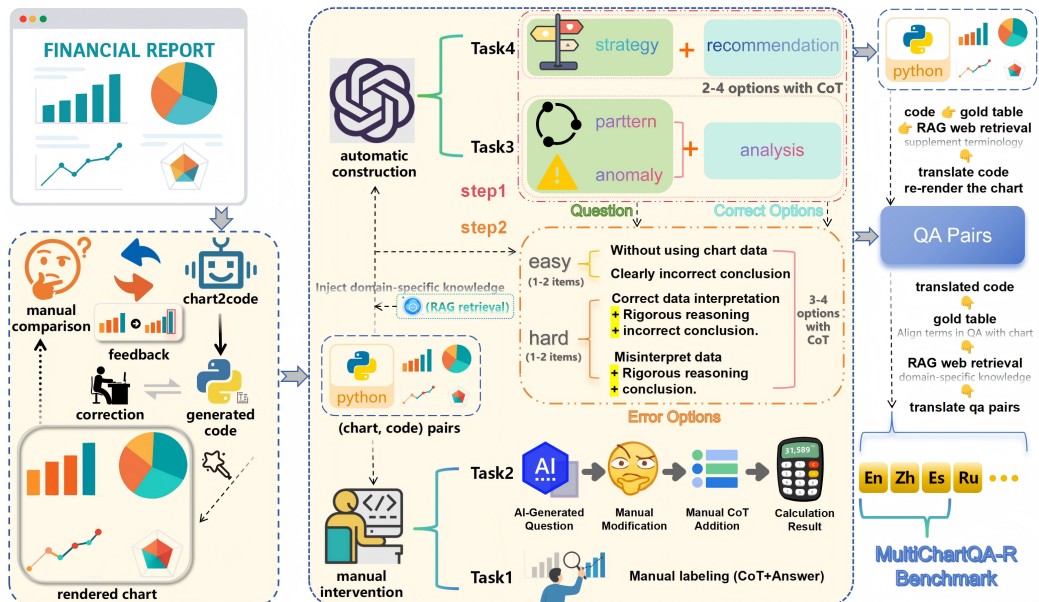

Figure 2: The Construction Pipeline of MultiChartQA-R

by combining chart information with relevant external knowledge. As shown in Task 3 of Figure 1, it involves both identifying surface-level data phenomena and investigating their underlying causes.

**Strategy Recommendation** Task 4 aims to evaluate the model's ability to extract relational patterns between different indicators (such as trade-offs, threshold critical points influencing decisions, etc.) based on multi-chart analysis results, and to propose optimization strategies or decision recommendations based on these patterns. As shown in Task 4 of Figure 1, the model must synthesize key information from multiple charts and clarify the interactions between indicators. The strategies generated should align with the patterns in the charts and have practical application value.

## 2.2 Construction Process

Figure 2 illustrates the construction process of MultiChartQA-R, from the collection of chart-code pairs, the construction of QA pairs, to the final multilingual expansion.

**Chart-Code Pairs Collection** We searched publicly available channels on the Internet for high-quality industry analysis reports and search-index dashboards, which often contain multiple interrelated charts for data analysis. Because the underlying raw data for these publicly shared charts is typically unavailable, we employed a human-in-the-loop process in which a large model reverses each chart into Python code. Through multiple rounds of manual interaction, we refined the generated code so that the reconstructed charts closely match the originals and preserve the conveyed information. This approach yields charts that reflect real-world patterns, conform to common sense, and carry greater value and significance, while human oversight ensures that the styles of the code-generated charts are attractive and diverse. We collected a total of 180 multi-chart sets and 695 chart-code pairs.

**Question–Answer Pair Construction** The first two tasks are manually annotated, with a standardized output format applied to the questions. For the latter two tasks, the synthesis process involves extracting a set of multi-chart gold tables from the code and using a reasoning model to generate questions, correct options, and the reasoning process for the correct options, with the task definition and gold tables serving as context in a few-shot manner. To further enhance the quality and relevance of the generated QA pairs, RAG web retrieval is employed to access domain-specific knowledge, supplementing the reasoning process with additional contextual information. The reasoning model is then used to generate 3-4 incorrect distractors, using the gold tables, correct question-answer pairs, and web-retrieved knowledge as context. These distractors are classified into two difficulty levels: "easy" and "hard." **Easy distractors** (1 or 2) do not rely on chart content and can be easily excluded based on general industry or logical knowledge, while **hard distractors** (1 or 2) involve misinter-

Table 1: Statistics of MultiChartQA-R

| Statistic | Number | Category | Number | Ans Type |
|---|---|---|---|---|
| Total Questions | 2160 | - Cross-chart Trend Inference | 540(25%) | Yes/No |
| Unique charts | 695 | - Complementary Data Integration | 540(25%) | Generation |
| Multi-chart sets | 180 | - Anomaly and Pattern Analysis | 540(25%) | Multi-option |
| Average charts | 3.9 | - Strategy Recommendation | 540(25%) | Multi-option |

preting the chart data or using rigorous reasoning that leads to an incorrect conclusion, making them harder to distinguish due to their more convincing structure and use of chart information.

**Multilingual Expansion** First, we construct charts in different languages by using LLMs to translate the textual information in the chart rendering code into the target languages, and then execute the code to render charts in those languages. During the code translation step, RAG is employed to search for relevant technical terms and specialized vocabulary to ensure accurate and contextually correct translation of the chart's content. Next, we construct question-answer pairs in different languages by extracting gold tables from the chart rendering code. The content of the gold tables is organized into text, describing each data point, and we feed these descriptions as context to the LLM during translation to ensure consistent terminology with the chart content. Additionally, RAG is utilized during QA pair translation to search for related domain-specific knowledge, further ensuring that the translation is consistent with the intended meaning and usage. We provide charts and QA pairs in English, Chinese, and Spanish, and this approach can be extended to additional languages to facilitate in-depth exploration of multilingual cross-chart tasks.

## 2.3 DATA STATISTICS & QUALITY INSPECTION

MultiChartQA-R includes 180 sets of charts and 695 chart-code pairs, covering three languages (extendable), as detailed in Table 1. In terms of chart types, it includes 14 categories of charts, spanning 36 domains, with specific details provided in Figures 4 and 3 in the appendix D.1.

We conducted a rigorous cross-review of all question–answer pairs in the first two tasks. Task 3 and Task 4 are multi-step synthesized data. We designed a supervised scoring mechanism to perform manual quality evaluation across three dimensions: question-type alignment, validity of correct options, and effectiveness of distractors. We divided the experts into a review group and a problem-solving group to score and solve a randomly sampled 30% of the data, respectively. The review group results showed that the average scores for both task types exceeded 9.0 (out of 10), while the problem-solving group achieved a 90+ $MF_\beta$ score and over 85% inter-rater consistency, indicating that the overall data quality is robust and reliable. This also reflects the rationality of the task design and its research value. Detailed evaluation metrics and processes can be found in the appendix D.2.

## 2.4 COMPARISONS WITH EXISTING BENCHMARKS

To further distinguish the difference between MultiChartQA-R and other existing ones, we elaborate the benchmark details in table 2. A comparison with other benchmarks clearly demonstrates that MultiChartQA-R excels in terms of broader scope, flexibility, and real-world applicability, offering superior quality and relevance for multi-chart question answering tasks.

Table 2: Comparison of MultiChartQA-R with existing chart-based QA benchmarks.

| Benchmarks | Reflect Real Scenarios | Topic Diversity | MultiChart | Multilingual | CoT | Chart-Code Pairs | Evaluation Metric | # of Chart Types |
|---|---|---|---|---|---|---|---|---|
| PlotQA (Methani et al., 2020) | ✗ | ✗ | ✗ | ✗ | ✗ | ✗ | Accuracy | 3 |
| ChartQA (Masry et al., 2022) | ✓ | - | ✗ | ✗ | ✗ | ✗ | Accuracy | 3 |
| ChartXiv (Wang et al., 2024) | ✓ | ✓ | ✗ | ✗ | ✗ | ✗ | GPT-4 Score | 18 |
| ChartQAPro (Masry et al., 2025) | ✓ | ✓ | ✓ | ✗ | ✗ | ✗ | Accuracy + ANLS score | 9+ |
| MultiChartQA (Zhu et al., 2025b) | ✓ | - | ✓ | ✗ | ✗ | ✗ | Accuracy | - |
| MultiChartQA-R(Ours) | ✓ | ✓ | ✓ | ✓ | ✓ | ✓ | Accuracy + $MF_\beta$-score | 14 |

## 3 EXPERIMENTS

## 3.1 EXPERIMENTAL SETUP

We conducted benchmark evaluations on 13 widely used proprietary and open-weight MLLMs in the field. For proprietary models, we selected three representative models: GPT-4o (OpenAI, 2024),

Claude-Sonnet-4 (Anthropic, 2025), and Gemini-2.5-Pro (Team, 2025), and also included a newly discovered proprietary model, Seed1.5-VL (Guo et al., 2025) in the evaluation. For open-weight MLLMs, we selected nine competitive models, with parameter sizes ranging from 7B to 78B: InternVL2(26B, 76B) (Chen et al., 2024b), InternVL3-78B (Zhu et al., 2025a), Qwen2.5-VL(7B, 72B) (Qwen et al., 2025), LLaVA-OV(7B, 72B) (Li et al., 2024a), DeepSeek-VL-7B (Lu et al., 2024), and MiniCPM-V-2.6 (Yao et al., 2024). All evaluations employed the Chain-of-Thought (CoT) (Wei et al., 2022) technique, and the corresponding prompts are provided in the appendix H.

## 3.2 EVALUATION METRIC

**Cross-Chart Trend Inference** employs strict string matching, as the questions explicitly constrain the answer format and are predominantly Yes/No judgments.

**Complementary Data Integration** addresses the fact that large language models struggle with arithmetic. We ask the model to extract the necessary values from the charts and outline the step-by-step reasoning. We then feed those steps into DeepSeek-V3.1 to generate executable Python code, run the code to obtain the final numeric result, and use a regular expression to extract the numeric component. We evaluate the correctness of the reasoning chain with a relaxed accuracy metric(Masry et al., 2022), thus testing the model's ability to perform long-form, multi-step inference.

**Anomaly and Pattern Analysis** and **Strategy Recommendation** involve open-ended multi-chart question-answering, where the answers are not unique, and human evaluators often make selections based on their preferences. Therefore, these tasks are designed in a multiple-choice format. To evaluate the model's performance, we propose an evaluation metric, the Multiple-choice $F_\beta$ Score ($MF_\beta$), which combines the base score, penalty, and a final composite score, comprehensively measuring the model's ability to balance correct selections and avoid incorrect ones. The process for constructing $MF_\beta$ is as follows.

*BaseScore* is used to assess the model's ability to select the correct answers. Let the set of correct answers be denoted as $A$ and the set of answers selected by the model as $B$. The BaseScore is defined as:

$$\text{BaseScore} = \frac{|A \cap B|}{|A|} \tag{1}$$

where $|A \cap B|$ is the number of correctly selected items by the model, and $|A|$ is the total number of correct answers. The BaseScore lies in the range $[0, 1]$, with a score of 1 for perfect correctness, a score between 0 and 1 for partial correctness, and a score of 0 for complete incorrectness.

*Penalty* measures the model's misselection behavior, especially when the model selects incorrect or interfering options. We classify interfering items into two categories: easy-to-confuse items (set $E$) and hard-to-confuse items (set $H$). Each time the model selects an interfering item, it incurs a penalty. The penalty coefficients $w_e$ and $w_h$ correspond to the easy and hard interfering items, respectively, and satisfy the constraint $w_e = 2w_h$ and $w_e \cdot |E| + w_h \cdot |H| = 1$, where $|E|$ and $|H|$ represent the number of easy and hard interfering items. The total penalty is then computed as:

$$\text{Penalty} = w_e \cdot |B \cap E| + w_h \cdot |B \cap H| \tag{2}$$

where $|B \cap E|$ and $|B \cap H|$ represent the numbers of easy and hard interfering items selected by the model.

If the score for each task is simply computed as $\text{Score} = \max(0, \text{BaseScore} - \text{Penalty})$, this formula ensures non-negative performance by considering both correct selections and error avoidance. However, a low score may indicate that the model is either too conservative (e.g., $|B| = 1$, $|E| = |H| = 0$) or too aggressive (e.g., $|B| = 4$, $|E| + |H| = 4$). To address this, we propose a more integrated metric that evaluates the BaseScore, Penalty, and Score simultaneously.

We introduce the $F_\beta$-score to multiple-choice tasks and construct the $MF_\beta$ evaluation metric, which considers two key aspects of performance: selecting correct answers (BasicScore) and avoiding incorrect ones (EscapeScore). This approach provides a more comprehensive assessment of the model's overall effectiveness.

$$\text{EscapeScore} = 1 - \text{Penalty} \tag{3}$$

$$MF_\beta = (1 + \beta^2) \times \frac{\text{BaseScore} \times \text{EscapeScore}}{\beta^2 \times \text{BaseScore} + \text{EscapeScore}} \tag{4}$$

where $\beta$ is a tuning parameter used to control the balance between BaseScore and EscapeScore. If $\beta = 1$, the model is required to both select correctly and avoid errors equally. When $\beta > 1$, greater emphasis is placed on avoiding incorrect selections, whereas if $\beta < 1$, the focus shifts toward selecting correct items.

This refined scoring mechanism offers a balanced approach for evaluating multi-option selection tasks by considering both the accuracy of selections and the avoidance of errors. We compare $MF_\beta$ with $Com^2$(Xiong et al., 2025) in appendix E.1, where we highlight $MF_\beta$'s role in model selection for specific scenarios and its feasibility for preference model training.

Table 3: The MultiChartQA-R leaderboard. The best scores are in bold.

| Model | Trend Inference | | | Data Integration | | | Anomaly/Pattern Attr | | | Strategy Rec | | |
|---|---|---|---|---|---|---|---|---|---|---|---|---|
| | en | zh | es | en | zh | es | en | zh | es | en | zh | es |
| Human | | 97.83 | | | 94.83 | | | 90.60 | | | 91.60 | |
| **Proprietary Models** | | | | | | | | | | | | |
| Claude-Sonnet-4 | 70.00 | 75.78 | 69.33 | 60.99 | 59.64 | 62.98 | **84.92** | 64.12 | **84.84** | **87.53** | 67.09 | **86.95** |
| Gemini-2.5-Pro | **75.06** | **79.33** | **75.11** | 65.08 | 69.35 | 65.91 | 81.87 | **83.92** | 82.48 | 83.79 | 84.49 | 83.96 |
| Seed1.5-VL | 72.44 | 71.78 | 67.33 | **67.66** | **72.81** | **68.64** | 78.87 | 82.46 | 78.69 | 82.22 | **85.38** | 78.81 |
| GPT-4o | 64.21 | 62.64 | 59.87 | 64.83 | 63.60 | 64.77 | 71.76 | 63.62 | 67.33 | 76.88 | 67.72 | 70.47 |
| **open-weight MLLMs** | | | | | | | | | | | | |
| InternVL3-78B | **73.21** | **68.46** | **64.66** | **67.50** | **70.72** | 63.33 | **81.62** | **82.22** | **78.16** | **81.78** | **84.48** | **76.57** |
| InternVL2-L3-76B | 59.91 | 48.88 | 59.19 | 51.59 | 53.51 | 51.14 | 68.33 | 71.69 | 70.86 | 70.19 | 76.51 | 75.77 |
| Qwen2.5-VL-72B | 56.25 | 56.95 | 53.13 | 25.40 | 14.77 | 21.51 | 72.62 | 75.89 | 71.12 | 76.34 | 78.75 | 72.85 |
| LLaVA-OV-72B | 61.33 | 57.59 | 53.33 | 43.02 | 16.25 | 22.22 | 66.82 | 66.82 | 66.58 | 71.37 | 68.62 | 69.79 |
| InternVL2-26B | 54.46 | 58.65 | 50.11 | 31.03 | 28.70 | 21.95 | 56.85 | 62.66 | 53.39 | 64.72 | 67.41 | 58.98 |
| Qwen2.5-VL-7B | 54.44 | 54.91 | 52.22 | 21.04 | 19.64 | 21.14 | 71.38 | 70.93 | 65.08 | 74.55 | 73.77 | 68.21 |
| MiniCPM-V-2_6 | 49.88 | 55.36 | 44.22 | 23.29 | 26.15 | 16.91 | 60.67 | 60.73 | 54.87 | 60.13 | 65.27 | 55.52 |
| DeepSeek-VL-7B | 49.32 | 47.11 | 40.44 | 7.95 | 5.01 | 6.74 | 49.90 | 48.30 | 48.30 | 56.95 | 52.51 | 49.91 |
| LLaVA-OV-7B | 48.55 | 34.00 | 46.00 | 21.84 | 8.20 | 12.38 | 52.18 | 52.40 | 56.14 | 58.98 | 62.31 | 61.30 |

## 3.3 MAIN RESULTS

Table 3 summarizes the evaluation results of 13 MLLMs on MultiChartQA-R. Our key observations are as follows:

**The foundational visual capabilities and data integration abilities of proprietary models show a significant gap compared to humans in cross-chart scenarios, but they have demonstrated good performance in pattern summarization and logical reasoning.** Among the proprietary models, Gemini-2.5-Pro shows superior trend-analysis ability across three languages. The newly released Seed1.5-VL achieves the best results on data integration task. InternVL3-78B exhibits outstanding performance, achieving state-of-the-art results among open-weight MLLMs and approaching the performance of proprietary models across all tasks.

**Proprietary models continue to outperform most open-weight MLLMs by a considerable margin.** Although InternVL3-78B achieves performance comparable to proprietary models, the remaining open-weight MLLMs lag substantially. This marked disparity confirms that MultiChartQA-R poses a significant challenge for current open-weight multimodal large language models. In particular, 7B and 8B open-weight MLLMs attain only near-random accuracy on trend-judgment task, revealing a lack of genuine trend-analysis capability, and their accuracy across the other three tasks also remains deficient. Nonetheless, a clear positive correlation is observed between parameter scale and performance on all four tasks. Overall, these results indicate that the open-source community still has ample scope to enhance MLLMs' competencies in complex visual understanding, cross-modal reasoning, conflict detection and attribution, and strategy formulation.

**MLLMs exhibit heterogeneous performance in multi-chart tasks across different languages.** Unlike other multilingual benchmarks, we did not observe English dominance. We speculate that the parameter subspace associated with data-analytic reasoning has limited overlap with that governing multilingual processing, thereby attenuating any potential English-language advantage. We will investigate MLLMs' performance in cross-lingual multi-chart tasks further in the Discussion section.

# 4 DISCUSSION

## 4.1 IMPACT OF IRRELEVANT CHARTS

During annotation, We recorded the specific charts associated with each QA pair in a "charts involved" field. In the main experiments, only these relevant charts were provided to the MLLMs. To investigate the impact of irrelevant ones on MLLMs' visual QA performance, we designed a comparative experiment in which additional unrelated charts were introduced alongside the relevant ones during inference. When the number of charts in a set is too large, the issue of excessively long visual tokens arises. The solution to this problem is provided in the appendix F.1.

Table 4 demonstrates that proprietary models maintain stable performance across the four tasks, even with additional chart inputs. This suggests that proprietary models possess strong capabilities for locating and retrieving relevant chart information when the question intent is clear. Open-weight MLLMs suffer performance drops across all four task categories, indicating that their chart-information retrieval and localization abilities still have substantial room for improvement.

## 4.2 PERFORMANCE ACROSS DIFFERENT LANGUAGES

To further assess multilingual performance in multi-chart QA, we conducted two comparative experiments. In the first setting, we used charts in English while varying the language of the question-answer pairs and prompts. In the second setting, the QA pairs and prompts remained in English, whereas the charts were translated into different languages. The results show that reasoning on English charts with prompts in different languages resulted in significant performance fluctuations. In contrast, when reasoning with different charts but the same language, the fluctuation in results across languages was smaller. This indicates that the cross-linguistic consistency of reasoning in multi-chart question answering tasks for MLLMs still requires improvement. The experimental data are presented in Table 6&7 of the appendix F.2.

## 4.3 EXPLORING MLLMS' RETRIEVAL CAPABILITIES

Additionally, we constructed a dedicated dataset to analyze large models' entity-extraction performance across multiple charts by extracting the numeric answers and computing relaxed accuracy.

Details about the extended-benchmark can be found in the appendix G. A brief overview of its four task types is as follows: Parallel-type question-answer pairs extract content from different charts based on independent sub-questions and list them individually. Union-type question-answer pairs extract content from different charts based on a single question, perform combination operations, and output a single answer. "PCPC" stands for "per chart per content" meaning each chart involves one piece of content. "PCMC" stands for "per chart multi-content" meaning each chart may involve more than one piece of content.

Comparing the results across the four sets in Figure 7 of appendix F.3, we observed that MLLMs' performance consistently worsens as the number of charts increases and the amount of information per chart grows. Interestingly, the experiments also revealed that when processing multiple charts—extracting one datum per chart—a simple additional computation step to produce a calculated result achieves a higher score than directly outputting multiple data points. This finding indicates that MLLMs still need to improve their ability to process multiple queries in parallel.

## 4.4 ERROR ANALYSIS

In the evaluation, GPT-4o exhibited a marked decline in accuracy in task 3 and task 4 multiple-choice tasks, with accuracy rates much lower than those of Claude-Sonnet-4 and Seed1.5-VL. We further discovered that this performance decline in GPT-4o is closely related to its tendency to mimic the format of one-shot example answers (B, C, E) in the prompt. Although the order of the options in the test questions had been shuffled, GPT-4o still frequently selected options B, C, and E. For example, in task 3, the misselection rates for B, C, and E were 48.0%, 53.4%, and 51.6%, respectively, significantly higher than the rates for other options (with an average misselection rate of 1.2% for other options) and other models (Claude had an average misselection rate of 10.4% for B, C, and E). This indicates that the model failed to reason based on the chart content and instead overly replicated

the structural pattern of the example answers, showing a strong dependency on example structure, which resulted in higher misselection rates and an overall performance decline.

In contrast, Claude-Sonnet-4 maintained a higher accuracy rate while demonstrating significantly lower misselection rates for non-example options (such as A, D, F), showing stronger suppression of prompt bias (Xu et al., 2024) (where pre-trained language models may develop unreasonable preferences for labels suggested by the prompt) and a more balanced judgment of the content of each option. Seed1.5-VL showed slightly lower accuracy, but its bias toward specific options was still noticeably better than that of GPT-4o. Overall, the models' performance on multiple-choice tasks is somewhat limited by their ability to suppress irrelevant structural information introduced by the prompt and to adapt to the actual semantic requirements of the questions. The statistical results are presented in the appendix F.4.

## 5 RELATED WORKS

**Chart Question Answering Benchmarks** Early benchmarks such as FigureQA (Kahou et al., 2018), DVQA (Kafle et al., 2018), PlotQA (Methani et al., 2020), and ChartQA (Masry et al., 2022) primarily focused on basic chart types, and addressed fundamental question-answering tasks such as data extraction. These tasks were limited in scope and did not fully cover the application of charts in complex and diverse environments. In recent years, with advancements in research, new benchmarks have emerged, such as ChartBench (Xu et al., 2023), ChartLlama (Han et al., 2023), Charxiv (Wang et al., 2024), and ChartAssistant (Meng et al., 2024), which enhance the diversity of both charts and questions. Additionally, tasks like Chart-to-code (Yang et al., 2024), which involve more challenging visual understanding, have also appeared. However, research on multi-chart question answering remains relatively scarce.

**Multilingual Chart Question Answering Benchmarks** The rapid growth of multilingual VQA benchmarks (e.g., xGQA (Pfeiffer et al., 2022), MaXM (Changpinyo et al., 2023), CVQA (Romero et al., 2025)) has addressed the English-centric bias in visual question answering. However, multilingual reasoning over structured charts remains severely . POLYCHARTQA (Xu et al., 2025) spans ten languages and reveals performance deficits on non-English inputs, but its tasks emphasize shallow extraction and lack true reasoning challenges. OneChart (Chen et al., 2024a)'s ChartY benchmark covers only Chinese and English and focuses on structural extraction, lacking a systematic evaluation of multilingual chart reasoning. KITAB-Bench (Heakl et al., 2025) targets English–Arabic chart localization but is limited both in language coverage and task depth.

**Multi-chart Question Answering Benchmarks** A series of multi-image question-answering benchmarks have emerged, such as Mantis-Instruct (Jiang et al., 2024), BLINK (Fu et al., 2024), and MUIRBENCH (Wang et al., 2025). However, these do not include chart-type images. MMC-Benchmark (Liu et al., 2024) contains a small subset of multi-chart data, but it only includes 52 samples. ReMI (Kazemi et al., 2024) includes some multi-chart scenarios, but the question types are limited. MultiChartQA (Zhu et al., 2025b) is the first benchmark specifically designed to explore multi-chart question answering, encompassing three types of multi-chart tasks: multi-chart information extraction, cross-chart data comparison, and sequential reasoning. While it reflects some of the capabilities of MLLMs in multi-chart reasoning tasks, it does not fully capture the real-world scenario demands. This paper introduces more realistic tasks that better reflect the performance aspects that are of greater concern to users.

## 6 CONCLUSION

In this paper, we introduce MultiChartQA-R, a benchmark designed to assess the multi-chart reasoning capabilities of MLLMs through four core tasks, each reflecting a crucial aspect of the multi-chart analytical reasoning process. Additionally, it can be extended to multiple languages. We also propose a flexible multiple-choice evaluation metric, $MF_\beta$, whose effectiveness is validated through formal reasoning and comparative experiments. Furthermore, we conduct extensive cross-chart question-answering and cross-language experiments on 13 mainstream MLLMs, revealing several intriguing phenomena. MultiChartQA-R serves as a foundation for advancing the development of more capable MLLMs in real-world scenarios.

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

APPENDIX

## A  ETHICS STATEMENT

This work adheres to the ICLR Code of Ethics. In this study, no animal experimentation was involved, and no personal data containing privacy or sensitive information was used.

Human annotators were engaged during dataset construction and evaluation. Their tasks were strictly limited to assessing the validity, logical consistency, and accuracy of chart-based question–answer pairs. These activities did not involve personally identifiable information and posed no privacy, safety, or psychological risks. All annotation and evaluation procedures were carried out under compliant and safe conditions.

We took care to avoid potential biases or discriminatory outcomes in both the dataset and the reported results. The authors are committed to maintaining transparency and academic integrity throughout the research process.

## B  REPRODUCIBILITY STATEMENT

We have made every effort to ensure that the results presented in this paper are reproducible. The paper provides detailed descriptions of the data annotation process, quality control mechanisms, experimental design, and evaluation methodology, enabling other researchers to understand and replicate our work. All code and datasets will be released in an anonymous repository upon publication to facilitate replication and verification. The comparative experiments reported in this paper are based on publicly available models and methods, ensuring consistent and reproducible evaluation results. We believe these measures will enable other researchers to reproduce our work and further advance the field.

## C  LLM USAGE

Large Language Models (LLMs) were partially used during this research. Specifically, in dataset construction, LLMs were employed in the initial generation of a subset of questions and answer options, after which human annotators conducted verification and quality checks to ensure accuracy and safety. In manuscript preparation, LLMs were used to assist with language polishing, improving clarity, accuracy, and overall fluency of the text.

It is important to note that LLMs were not involved in research ideation, methodological design, or experimental planning. All research concepts, scientific claims, and data analyses were independently developed and carried out by the authors. The authors take full responsibility for the content of the manuscript, including sections that involved LLM assistance, and have ensured that the use of LLMs complies with academic ethical standards without contributing to plagiarism or research misconduct.

## D  BENCHMARK

### D.1  STATISTICS

The statistical results can be found in Figure 3 and Figure 4.

### D.2  QUALITY INSPECTION

For the first two task types—true/false and numerical-answer questions, both characterized by objectively verifiable answers—we implemented a systematic cross-review of all question–answer pairs. The annotation team consisted of four members and employed a cyclic peer-review mechanism. Each annotator's work was independently verified by another member, and any identified errors were promptly corrected. The review criteria included:

- **Question validity**: ensuring that each question conformed to its definition and was clearly formulated;

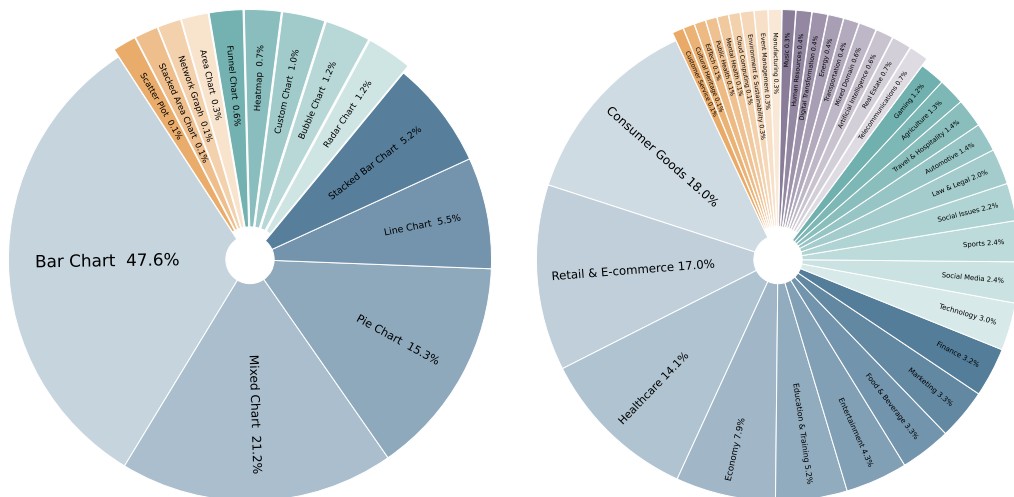

Figure 3: Distribution of Domains      Figure 4: Distribution of Chart Types

- **Reasoning correctness**: verifying that the annotated reasoning chain was logically sound and rigorous;
- **Answer accuracy**: confirming that the final answer was fully consistent with the chart data.

Through this process, we ensured the dataset's robust quality and reliability in terms of question formulation, reasoning, and answer accuracy.

For the two more challenging task types, we randomly sampled 30% of the data for human quality evaluation. Each question–answer pair was scored by evaluators with reference to the reasoning chain used during its generation. We applied a 10-point evaluation scale based on three criteria:

- **Question-type alignment**: assessing whether the question corresponded to the intended task type and was reasonably designed;
- **Validity of correct options**: ensuring that the reasoning behind correct options was strictly grounded in the chart data, logically coherent, and led to reliable conclusions;
- **Effectiveness of distractors**: requiring simple distractors to appear superficially plausible yet independent of the chart data, and difficult distractors to superficially rely on chart reasoning while containing critical logical flaws (e.g., misinterpreting a downward trend as upward).

Evaluation results indicate that the average score for the third task type was 9.1/10, with an inter-rater agreement of 85%, while the fourth task type achieved an average score of 9.3/10 and an inter-rater agreement of 87%, demonstrating highly robust overall evaluation outcomes.

In addition, the evaluators completed all four task types, obtaining corresponding scores of 97.83, 94.83, 90.60, and 91.60, respectively. Human performance consistently exceeded that of the models, though a non-negligible error rate remained. This observation both confirms that the tasks remain challenging for current models and highlights the intrinsic difficulty of the tasks, underscoring their research value.

## E  EXPERIMENTS

### E.1  EVALUATION METRIC

$MF_\beta$ and $Com^2$ exhibit similar distributions across the evaluation of various models, indicating that $MF_\beta$ can effectively capture the model's multiple-choice capabilities. However, $Com^2$ amplifies

the impact of distractors, making its evaluation results not applicable in all scenarios. In contrast, $MF_\beta$ can be more flexible and applicable to a wider range of situations by adjusting the $\beta$ parameter.

### ANALYSIS OF $MF_\beta$ CURVES

We plotted the $MF_\beta$ curves of all models under varying values of $\beta$. Overall, the curves exhibit a monotonically increasing trend, indicating that within the current task setting, "selecting all correct options" is significantly more difficult than "avoiding incorrect options." In other words, the models generally perform better at eliminating incorrect options than at fully covering the correct ones. We hypothesize that this phenomenon is related to the characteristics of the hard options: such options are more prone to being selected by the models, yet their penalty weight in the scoring scheme is relatively low, which to some extent "inflates" the overall scores. This effect is particularly pronounced for smaller-parameter models whose performance approaches random selection.

**Intersection points.** The intersections between the curves carry critical implications: they reveal the trade-offs between different models in terms of "recalling correct options" versus "avoiding incorrect ones," thereby providing guidance for model selection across application scenarios. For example, in recall-oriented tasks, models that perform better before the intersection point should be prioritized, whereas in precision-oriented tasks, models that excel after the intersection point are preferable.

**Curve variability.** Taking InternVL2-26B as an example, its curve ranks relatively low when $\beta$ is small, but improves markedly as $\beta$ increases. This pronounced change highlights the model's substantial variability across different evaluation emphases, reflecting a lack of balance—that is, an insufficiently stable ability to reconcile recall and precision.

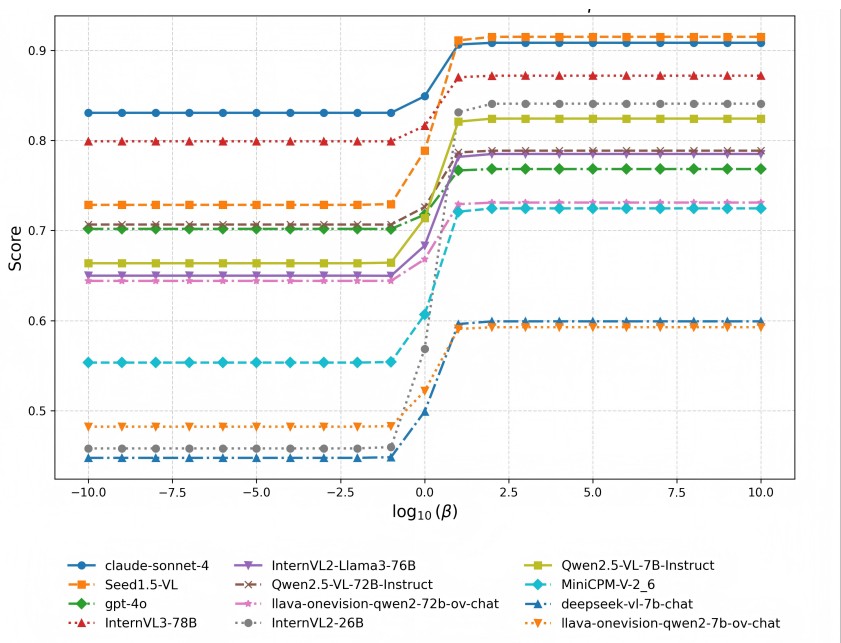

Figure 5: Model Performance on Task 3 under Different $\beta$

## F   DISCUSSION

### F.1   IMPACT OF IRRELEVANT CHARTS

In these experiments, because the number of charts varies and high-quality charts with many visual tokens can cause the total input length to exceed the model's context window, the model will begin

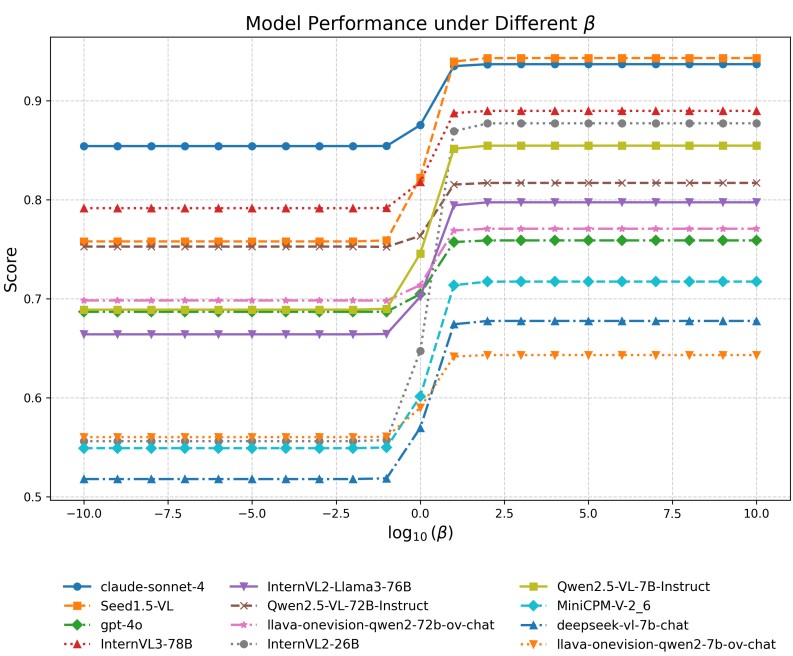

Figure 6: Model Performance on Task 4 under Different $\beta$

discarding tokens from the left (i.e., the earliest inputs) until the remaining token count less than session_len. Therefore, we deliberately placed any image tokens for charts unrelated to the QA pair on the far left; even if they are dropped, the model's ability to answer correctly remains unaffected.

**Method for handling excessively long image tokens.** In these experiments, because the number of charts varies and high-quality charts with many visual tokens can cause the total input length to exceed the model's context window, the model will begin discarding tokens from the left (i.e., the earliest inputs) until the remaining token count less than session_len. Therefore, we deliberately placed any image tokens for charts unrelated to the QA pair on the far left; even if they are dropped, the model's ability to answer correctly remains unaffected.

Table 4: Compare the question-answering performance when inputting all charts versus only the relevant charts.

| Model | Trend Inference | | Data Integration | | Anomaly/Pattern Attr | | Strategy Rec | |
|---|---|---|---|---|---|---|---|---|
| | involved | all | involved | all | involved | all | involved | all |
| Claude-Sonnet-4 | 70.00 | 69.11 | 60.99 | 61.38 | 84.92 | 85.32 | 87.53 | 87.85 |
| Seed1.5-VL | 72.44 | 72.83 | 67.66 | 66.74 | 78.87 | 79.08 | 82.22 | 81.95 |
| GPT-4o | 64.21 | 66.14 | 64.83 | 63.10 | 71.76 | 70.79 | 76.88 | 76.71 |
| InternVL3-78B | 73.21 | 60.93 | 67.50 | 64.68 | 81.62 | 81.56 | 81.78 | 80.57 |
| InternVL2-Llama3-76B | 59.91 | 52.46 | 51.59 | 44.36 | 68.33 | 67.23 | 70.19 | 70.04 |
| Qwen2.5-VL-72B-Instruct | 56.25 | 58.93 | 25.40 | 22.94 | 72.62 | 73.48 | 76.34 | 74.34 |
| LLaVA-OV-72B | 61.33 | 59.11 | 43.02 | 38.07 | 66.82 | 65.47 | 71.37 | 70.69 |
| InternVL2-26B | 54.46 | 46.33 | 31.03 | 22.90 | 56.85 | 54.53 | 64.72 | 64.37 |
| Qwen2.5-VL-7B-Instruct | 44.00 | 53.56 | 21.04 | 23.98 | 71.38 | 66.70 | 74.55 | 71.51 |
| MiniCPM-V-2_6 | 49.88 | 52.89 | 23.29 | 27.48 | 60.67 | 59.85 | 60.13 | 59.44 |
| DeepSeek-VL-7B | 49.32 | 24.22 | 7.95 | 3.48 | 49.90 | 45.58 | 56.95 | 55.74 |
| LLaVA-OV-7B | 48.55 | 45.19 | 21.84 | 13.55 | 52.18 | 52.10 | 58.98 | 57.92 |

Table 5: Compare the question-answering performance when inputting all charts versus only the relevant charts under the com2 metric.

| Model | Trend Inference | | Data Integration | | Anomaly/Pattern Attr | | Strategy Rec | |
|---|---|---|---|---|---|---|---|---|
| | involved | all | involved | all | involved | all | involved | all |
| Claude-Sonnet-4 | 70.00 | 69.11 | 60.99 | 61.38 | 65.04 | 62.56 | 69.67 | 69.15 |
| Seed1.5-VL | 72.44 | 72.83 | 67.66 | 66.74 | 58.72 | 58.63 | 66.30 | 63.81 |
| GPT-4o | 64.21 | 66.14 | 64.83 | 63.10 | 39.29 | 35.55 | 39.13 | 38.54 |
| InternVL3-78B | 73.21 | 60.93 | 67.50 | 64.68 | 57.37 | 54.25 | 57.20 | 54.20 |
| InternVL2-Llama3-76B | 59.91 | 52.46 | 51.59 | 44.36 | 34.24 | 33.99 | 36.95 | 36.18 |
| Qwen2.5-VL-72B-Instruct | 56.25 | 58.93 | 25.40 | 22.94 | 32.93 | 33.19 | 39.44 | 37.26 |
| LLaVA-OV-72B | 61.33 | 59.11 | 43.02 | 38.07 | 27.26 | 27.48 | 31.67 | 30.91 |
| InternVL2-26B | 54.46 | 46.33 | 31.03 | 22.90 | 36.32 | 30.81 | 36.53 | 37.27 |
| Qwen2.5-VL-7B-Instruct | 44.00 | 53.56 | 21.04 | 23.98 | 37.33 | 32.81 | 43.19 | 39.52 |
| MiniCPM-V-2_6 | 49.88 | 52.89 | 23.29 | 27.48 | 21.27 | 24.63 | 21.90 | 22.86 |
| DeepSeek-VL-7B | 49.32 | 24.22 | 7.95 | 3.48 | 11.35 | 7.59 | 17.19 | 16.30 |
| LLaVA-OV-7B | 48.55 | 45.19 | 21.84 | 13.55 | 9.20 | 8.97 | 12.03 | 10.13 |

## F.2 PERFORMANCE ACROSS DIFFERENT LANGUAGES

Table 6: English Charts - Different Language QAs

| Model | Trend Inference | | | Data Integration | | | Anomaly/Pattern Attr | | | Strategy Rec | | |
|---|---|---|---|---|---|---|---|---|---|---|---|---|
| | en | zh | es | en | zh | es | en | zh | es | en | zh | es |
| Seed1.5-VL | 72.44 | 71.56 | 67.56 | 67.66 | 68.76 | 67.04 | 78.87 | 81.05 | 79.22 | 82.22 | 85.17 | 79.19 |
| InternVL3-78B | 73.21 | 70.98 | 66.00 | 67.50 | 70.69 | 65.91 | 81.62 | 83.47 | 78.54 | 81.78 | 84.07 | 76.91 |
| InternVL2-26B | 54.46 | 46.09 | 45.07 | 31.03 | 28.57 | 21.72 | 56.85 | 61.36 | 56.91 | 64.72 | 66.25 | 60.77 |
| MiniCPM-V-2_6 | 49.88 | 51.79 | 48.78 | 23.29 | 29.50 | 23.95 | 60.67 | 63.64 | 56.57 | 60.13 | 66.14 | 59.62 |
| Qwen2.5-VL-7B | 54.44 | 55.23 | 48.67 | 21.04 | 23.81 | 23.29 | 49.90 | 70.74 | 66.25 | 56.95 | 72.34 | 68.45 |
| LLaVA-OV-7B | 48.55 | 33.48 | 46.33 | 21.84 | 18.26 | 13.14 | 52.18 | 54.10 | 59.92 | 58.98 | 66.08 | 62.03 |

Table 7: Different Language Charts - English QAs

| Model | Trend Inference | | | Data Integration | | | Anomaly/Pattern Attr | | | Strategy Rec | | |
|---|---|---|---|---|---|---|---|---|---|---|---|---|
| | en | zh | es | en | zh | es | en | zh | es | en | zh | es |
| Seed1.5-VL | 72.44 | 68.22 | 69.78 | 67.66 | 65.69 | 65.70 | 78.87 | 80.26 | 80.66 | 82.22 | 82.78 | 81.30 |
| InternVL3-78B | 73.21 | 68.97 | 69.87 | 67.50 | 67.26 | 58.03 | 81.62 | 80.88 | 80.93 | 81.78 | 82.46 | 81.97 |
| InternVL2-26B | 54.46 | 53.56 | 48.44 | 31.03 | 18.20 | 19.46 | 56.85 | 56.36 | 53.09 | 64.72 | 65.30 | 61.48 |
| MiniCPM-V-2_6 | 49.88 | 50.00 | 49.56 | 23.29 | 24.53 | 25.29 | 60.67 | 61.81 | 61.81 | 60.13 | 62.21 | 61.29 |
| Qwen2.5-VL-7B | 54.44 | 52.35 | 52.67 | 21.04 | 14.41 | 20.95 | 49.90 | 70.11 | 72.74 | 56.95 | 72.65 | 74.33 |
| LLaVA-OV-7B | 48.55 | 41.67 | 46.85 | 21.84 | 5.11 | 14.25 | 52.18 | 52.03 | 52.95 | 58.98 | 55.96 | 57.67 |

Table 8: The test results of English Q&A on mixed-language charts.

| Model | Trend Inference | Data Integration | Anomaly/Pattern Attr | Strategy Rec |
|---|---|---|---|---|
| Seed1.5-VL | 70.38 | 65.11 | 80.45 | 82.83 |
| InternVL3-78B | 69.93 | 64.96 | 81.78 | 81.75 |
| Qwen2.5-VL-7B-Instruct | 54.12 | 19.38 | 70.76 | 72.71 |

Table 9: English Charts - Different Language QAs Under the Com2 Metric

| Model | Trend Inference | | | Data Integration | | | Anomaly/Pattern Attr | | | Strategy Rec | | |
|---|---|---|---|---|---|---|---|---|---|---|---|---|
| | en | zh | es | en | zh | es | en | zh | es | en | zh | es |
| Seed1.5-VL | 72.44 | 71.56 | 67.56 | 67.66 | 68.76 | 67.04 | 58.72 | 63.57 | 59.39 | 66.30 | 69.63 | 62.52 |
| InternVL3-78B | 73.21 | 70.98 | 66.00 | 67.50 | 70.69 | 65.91 | 57.15 | 55.21 | 51.69 | 57.20 | 58.60 | 52.78 |
| InternVL2-26B | 54.46 | 46.09 | 45.07 | 31.03 | 28.57 | 21.72 | 36.32 | 26.54 | 26.76 | 36.53 | 31.22 | 29.44 |
| MiniCPM-V-2_6 | 49.88 | 51.79 | 48.78 | 23.29 | 29.50 | 23.95 | 21.27 | 25.93 | 17.44 | 21.90 | 24.93 | 18.01 |
| Qwen2.5-VL-7B | 54.44 | 55.23 | 48.67 | 21.04 | 23.81 | 23.29 | 37.33 | 35.05 | 39.07 | 43.19 | 36.07 | 48.67 |
| LLaVA-OV-7B | 48.55 | 33.48 | 46.33 | 21.84 | 18.26 | 13.14 | 9.20 | 11.19 | 18.52 | 12.03 | 20.04 | 17.04 |

Table 10: Different Language Charts - English QAs Under the Com2 Metric

| Model | Trend Inference | | | Data Integration | | | Anomaly/Pattern Attr | | | Strategy Rec | | |
|---|---|---|---|---|---|---|---|---|---|---|---|---|
| | en | zh | es | en | zh | es | en | zh | es | en | zh | es |
| Seed1.5-VL | 72.44 | 68.22 | 69.78 | 67.66 | 65.69 | 65.70 | 58.72 | 63.06 | 61.43 | 66.30 | 66.00 | 63.52 |
| InternVL3-78B | 73.21 | 68.97 | 69.87 | 67.50 | 67.26 | 58.03 | 57.15 | 56.34 | 53.75 | 57.20 | 55.47 | 57.23 |
| InternVL2-26B | 54.46 | 53.56 | 48.44 | 31.03 | 18.20 | 19.46 | 36.32 | 35.51 | 30.54 | 36.53 | 41.07 | 35.94 |
| MiniCPM-V-2.6 | 49.88 | 50.00 | 49.56 | 23.29 | 24.53 | 25.29 | 21.27 | 23.19 | 23.04 | 21.90 | 21.86 | 21.41 |
| Qwen2.5-VL-7B | 54.44 | 52.35 | 52.67 | 21.04 | 14.41 | 20.95 | 37.33 | 35.26 | 41.41 | 43.19 | 39.20 | 43.52 |
| LLaVA-OV-7B | 48.55 | 41.67 | 46.85 | 21.84 | 5.11 | 14.25 | 9.20 | 8.43 | 8.75 | 12.03 | 8.48 | 11.15 |

## F.3 EXPLORING MLLMS' RETRIEVAL CAPABILITIES

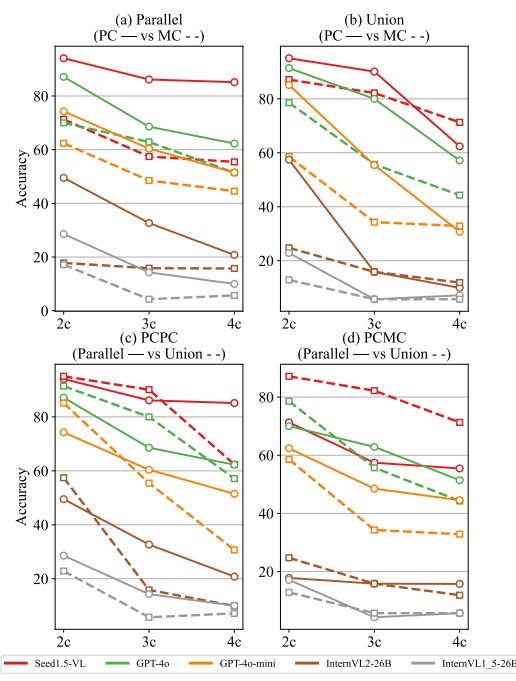

Figure 7: Compare the performance of MLLMs across four retrieval tasks. The x-axis represents the number of charts involved in the question, for example, 2c denotes two charts.

## F.4 ERROR ANALYSIS

The detailed response statistics of the four proprietary models on Task 3 and Task 4 are presented in Tables 11 & 12.

Table 11: Omission and multiple-selection rates per option for each model for task 3 (%). In this context, "claude" refers to Claude-Sonnet-4, "seed" refers to Seed1.5-VL, "gpt" refers to GPT-4o, and "gemini" refers to Gemini-2.5-Pro.

| Option | claude | seed | gpt | gemini |
|---|---|---|---|---|
| A | 24.5 / 6.3 | 36.8 / 0.8 | 64.2 / 1.3 | 26.9 / 1.7 |
| B | 7.9 / 9.2 | 10.2 / 8.1 | 3.4 / 48.0 | 2.3 / 19.0 |
| C | 9.6 / 10.3 | 6.9 / 12.6 | 4.8 / 53.4 | 2.7 / 24.4 |
| D | 17.3 / 6.6 | 36.1 / 0.8 | 76.0 / 0.8 | 36.1 / 1.2 |
| E | 9.4 / 11.6 | 9.9 / 10.5 | 4.7 / 51.6 | 1.6 / 23.3 |
| F | 8.9 / 5.6 | 29.7 / 1.6 | 63.4 / 1.6 | 34.7 / 1.6 |
| G | 12.4 / 4.3 | 33.8 / 1.6 | 57.9 / 1.0 | 26.9 / 0.7 |

Table 12: Omission and multiple-selection rates per option for each model for task 4 (%). In this context, "claude" refers to Claude-Sonnet-4, "seed" refers to Seed1.5-VL, "gpt" refers to GPT-4o, and "gemini" refers to Gemini-2.5-Pro.

| Option | claude | seed | gpt | gemini |
|--------|--------|------|-----|--------|
| A | 11.6 / 4.4 | 40.7 / 1.2 | 62.8 / 0.4 | 31.6 / 0.5 |
| B | 4.3 / 9.1 | 2.9 / 7.9 | 1.9 / 49.2 | 3.0 / 21.1 |
| C | 2.5 / 6.8 | 2.5 / 10.4 | 1.0 / 47.4 | 1.2 / 30.5 |
| D | 8.9 / 2.7 | 32.8 / 0.0 | 62.0 / 1.2 | 25.2 / 0.9 |
| E | 4.2 / 11.6 | 7.8 / 12.8 | 2.1 / 58.1 | 0.0 / 32.7 |
| F | 9.1 / 4.4 | 25.6 / 0.7 | 58.0 / 2.9 | 26.3 / 3.8 |
| G | 7.3 / 4.0 | 35.8 / 0.0 | 58.9 / 2.3 | 34.2 / 0.0 |

# G EXTENDED BENCHMARK.

## G.1 TASK DESCRIPTION

PARALLEL TYPE

- EXAMPLE
  **question:** question_1 about chart_1 question_2 about chart_2 question_3 about chart_3 question_4 about chart_4
  **answer:** answer1. answer2. answer3. answer4.

- PCPC
  A question contains multiple parallel sub-questions, each retrieving one piece of information from a single chart.
  2c, 3c, and 4c refer to retrieving one piece of information from charts 2, 3, and 4, respectively, and providing separate answers.

- PCMC
  A question contains multiple parallel sub-questions, each retrieving multiple pieces of information from a single chart.
  2c, 3c, and 4c refer to retrieving at least one piece of information from charts 2, 3, and 4, respectively, and providing separate answers.

UNION TYPE

- EXAMPLE
  **question:** question about chart_1&2&3&4
  **answer:** answer.

- PCPC
  The question involves multiple charts, retrieving one piece of information from each chart, combining the information to perform a simple calculation, and outputting the final result.

- PCMC
  The question involves multiple charts, retrieving at least one piece of information from each chart, combining the information to perform a simple calculation, and outputting the final result.

## G.2 PIPELINE

1. Manually select articles from Pew that contain at least four charts, and scrape both the articles and charts.

2. Extract chart information and perform manual sampling-based screening to ensure quality.

3. Automatically generate topic summaries based on the content of the article.

4. Generate question-and-answer pairs by utilizing the extracted chart information and setting different prompts based on the topics.

5. Use scripts to check whether the number of charts involved in all question-and-answer pairs and the amount of content related to each chart meet the expected criteria. If they do not meet the criteria, regenerate the pairs and manually adjust them until all question-and-answer pairs satisfy the conditions.

6. Use scripts to check the length of all answers, manually screen answers that exceed the threshold, remove redundant parts, and retain concise answers.

7. Utilize the rationale of generated question-and-answer pairs to generate code-based computation results, manually compare all inconsistent answers, correct the answers and rationales in the labels, and ensure the accuracy of the computation results.

## G.3 DATA STATISTICS

We collected 101 articles from Pew Research Center, each containing at least four charts centered on the same topic, spanning nine topics in total (Refer to Table 13), yielding 1,212 QA pairs. These QA pairs span four distinct question types, comprehensively probing models' capabilities in cross-chart information retrieval.

Table 13: Category-wise Number of Articles

| Category | Number of Articles |
|---|---|
| Economy & Work | 11 |
| Politics & Policy | 11 |
| Internet & Technology | 11 |
| Family & Relationships | 11 |
| Age & Generations | 11 |
| Immigration & Migration | 11 |
| Science | 11 |
| News Habits & Media | 12 |
| Other Topics | 12 |

# H  QAS PROMPTS

**Task 1**

```
output_prompt = '''\nOutput according to the following json format:
{
    "rationale": "Reasoning process",
    "answer": "Final answer"
}\nPlease strictly output according to the json format.\n'''

prompt = image_tokens + question + output_prompt
```

**Task 2**

```
output_prompt = '''
Please answer according to the following steps, **all details must
    be included and must not be omitted**:
1. Extract relevant information: List all data required for the
    calculation and indicate which chart each data comes from.
2. Explain the association logic: Explain why these data are needed
    .
3. Calculation process: Write out the detailed calculation process.
4. Conclusion summary: Answer according to the format required in
    the question.
5. Output requirements: Extract the output format requirements
    after the question.\n'''

prompt = image_tokens + question + output_prompt
```

**Task 3**

```
text1 = "The options for the question are as follows: "
hints = "(Multiple choice question) Please analyze the content of
    the chart and select the correct options based on the chart
    information. Note: Correct options should be supported by chart
    data; otherwise, they will be regarded as incorrect."
output_prompt = '''\nAnswer the question according to the following
     json format:
{
    "rationale": "Reasoning process",
    "answer": ["B", "C", "E"] /* Select the correct one or more
        options according to the actual situation */
}\n'''

prompt = image_tokens + hints + question + text1 + answer_choices +
     output_prompt
```

**Task 4**

```
text1 = "The options for the question are as follows: "
hints = "(Multiple choice question) Please analyze the content of
    the chart and select the correct options based on the chart
    information. Note: Correct options should be supported by chart
    data; otherwise, they will be regarded as incorrect."
output_prompt = '''\nAnswer the question according to the following
     json format:
```

```
{
    "rationale": "Reasoning process",
    "answer": ["B", "C", "E"] /* Select the correct one or more
        options according to the actual situation */
}\n'''

prompt = image_tokens + hints + question + text1 + answer_choices +
    output_prompt
```

**The prompt for generating python calculation code based on the reasoning process in task 2 is as follows:**

```
gen_code_prompt = '''
The above is the reasoning process. Since large models are not good
    at calculations, ignore the calculated results in the reasoning
    process above, and generate executable Python code for the
    reasoning process.
Please note:
1. Strictly generate Python code based on the "rationale" process
    above.
2. Ensure the code correctly reflects the reasoning of the "
    rationale" by using variables to replace the intermediate
    calculation results in the rationale, as the code execution is
    more accurate and avoids cumulative errors caused by using
    intermediate calculation results.
3. The final output should strictly follow the required format,
    only printing the last output from the final `print` statement,
    without any descriptive print statements. Below are some
    examples of the final print outputs for reference:
    a. Output format: "The answer should be presented as an integer
        ."
        Incorrect final output 1: print(f"The final answer is: {
            answer}.")
        Incorrect final output 2: print(f"**{answer}**")
        Correct final output: print(f"{answer}")
    b. Output in percentage, rounded to 4 significant digits.
        Incorrect final output 1: print(f"In mobile search, the
            proportion of images is **{answer}%**.")
        Correct final output: print(f"{answer}%")
    c. Output in dollars, rounded to 3 decimal places. Answer: xx
        dollars.
        Incorrect final output 1: print(f"In Q1 2023, the in-app
            purchase revenue per download for mobile games on the
            App Store in Japan is {answer} USD.")
        Correct final output: print(f"{answer} USD")
    d. Output "Yes" or "No".
        Incorrect final output 1: print("True")
        Correct final output: print("Yes")
    e. Answer: xx times.
        Incorrect final output 1: print(f"{answer}")
        Correct final output: print(f"{answer} times")
4. Be sure to distinguish between "significant digits" and "decimal
    places." "Significant digits" refer to all digits from the
    first non-zero digit to the last digit. "Decimal places" refer
    to the number of digits after the decimal point.
5. If floating-point calculations are involved, please use the `
    decimal` library to avoid precision loss from floating-point
    operations.
'''
```

# I TASK 3&4 SYNTHESIS METHOD

```
Algorithm 1: QA_Pair_Synthesis

Input:
    C <= Original code rendering scripts for multiple charts
    Q_defs <= Definitions of question types (templates + few-shot
        examples)
    M <= Reasoning model with chain-of-thought capabilities
Output:
    QA_dataset <= Complete set of question-answer pairs (including
        correct options and distractors)

1.  // Step 1: Extract Gold Tables
2.  gold_tables <= ExtractTablesFromCode(C)

3.  // Step 2: Generate questions and correct answers
4.  for each table_set in gold_tables:
5.      context <= Concatenate(Q_defs, table_set)
6.      prompt_correct <= BuildPrompt(context, task="generate
    question + correct answer + reasoning")
7.      (Q, A_correct, Reasoning) <= M.generate(prompt_correct)
8.      store CorrectPair = (Q, A_correct, Reasoning)

9.  // Step 3: Generate Easy and Hard distractors
10. for each CorrectPair in dataset:
11.     // Easy distractors
12.     prompt_easy <= BuildPrompt(
13.         context=(table_set, CorrectPair),
14.         instructions="generate 1 or 2 easy distractors;
    responses that do not include information from the table, only
    based on internal knowledge"
15.     )
16.     D_easy <= M.generate(prompt_easy, count=2)

17.     // Hard distractors
18.     prompt_hard <= BuildPrompt(
19.         context=(table_set, CorrectPair),
20.         instructions="generate 1 or 2 hard distractors;
    logically or numerically incorrect but superficially dependent
    on the table"
21.     )
22.     D_hard <= M.generate(prompt_hard, count=2)

23.     // Merge and shuffle all options
24.     all_choices <= [A_correct] U D_easy U D_hard
25.     random.shuffle(all_choices)  // Shuffle all options
    randomly

26.     // Label the correct option after shuffling
27.     correct_option <= all_choices.index(A_correct)  // Find the
     position of the correct option after shuffling

28.     // Add the question, shuffled options, and correct option
    to the dataset
29.     QA_dataset.add( Q, all_choices, correct_option )

30. return QA_dataset
```