# OpenReview forum: "MultiChartQA-R: A Benchmark for Multi-Chart Question Answering in Real-World Reasoning Scenarios"
_ICLR.cc/2026/Conference — ICLR 2026 Conference Withdrawn Submission_

### Official Review · Reviewer_5FDr · 2025-10-25

**Soundness:** 2
**Presentation:** 3
**Contribution:** 2
**Rating:** 4
**Confidence:** 5

**Summary:**

The paper proposes MultiChartQA-R, aiming to evaluate question-answering and reasoning abilities in multi-chart scenarios. It covers four progressively complex tasks: cross-chart trend judgment, complementary data integration, anomaly and causal/pattern analysis, and strategy recommendation. The dataset reportedly includes English, Chinese, and Spanish versions, and introduces a tunable multiple-choice evaluation metric, MFβ, to balance “accuracy” and “error avoidance.” The authors evaluate 13 multimodal large models and compare them with human performance, claiming that models show a significant gap from humans in cross-chart perception and data integration, as well as certain multilingual differences. The data are derived from multiple charts in public industry reports or dashboards, reconstructed via “reverse chart-to-code” generation and further verified through human review and RAG-based correction. Tasks 3 and 4 are model-generated and human-refined. The paper also presents overall statistics and comparison tables with existing benchmarks.

**Strengths:**

1. Starting from the four-step chain of “correlation → utilization → depth → application,” the paper decomposes multi-chart linkage analysis common in real business scenarios into task categories. The task design has some intuitive rationale, especially in isolating “strategy recommendation” as a decision-oriented task that helps capture application scenarios closer to end-user needs.
2. By translating rendering code terminology and aligning it with RAG terminology, the authors generate multilingual chart versions and perform consistent QA translation. The process is clearly described and can be extended to additional languages.
3. The benchmark covers 13 models (both proprietary and open-source, 7B–78B), reports multilingual results, and includes ablation analyses such as “irrelevant chart interference” and “expanding number of charts,” providing reference value.

**Weaknesses:**

1. The paper acknowledges the existence of prior multi-chart and multi-image reasoning benchmarks (e.g., MultiChartQA). Its main additions are the “strategy recommendation task” and multilingual support, but it lacks strong evidence that these four task types are inherently closer to real-world needs or that strategy recommendation remains truly open-ended after being formatted as multiple-choice. Although the comparison table claims broader coverage and more flexible evaluation, no controlled experiment is provided to demonstrate empirical improvement over MultiChartQA under identical models and question types.
2. The abstract and Table 1 give inconsistent impressions about whether “each language includes 695 charts and 2160 QA pairs” or whether these totals refer to the entire dataset. The abstract states that “each language version contains 695 chart-code pairs and 2160 QA,” while Table 1 lists “Total Questions 2160, Unique Charts 695” without clarifying whether this is per-language. The relationship between per-language and overall counts should be clearly unified and specified.
3. Tasks 3 and 4 are generated through a multi-step process of “extracting gold charts → model-generated questions and correct/distractor options → human review,” enhanced by retrieved external knowledge for reasoning. If the generation model shares data or stylistic overlap with the evaluated models, it may introduce “style leakage” or “distractor distribution bias.” The paper only reports high human quality scores but lacks quantitative comparisons to assess such risks (e.g., cross-model/cross-family stability, or how different generators affect task difficulty).
4. The paper defines constraints w_e = 2w_h and w_e|E| + w_h|H| = 1 but does not explain the source of these coefficients, nor provide sensitivity analysis or comparative justification (e.g., whether different w_e:w_h ratios produce stable model rankings or could be data-driven). Comparisons with existing multiple-choice metrics (e.g., weighted F1, Com2) are mostly relegated to the appendix, leaving the main text short of empirical validation.
5. Table 3 reports multi-model results but lacks statistical significance tests, confidence intervals, or repetition counts. The conclusion that “English has no advantage” is a strong inference requiring stricter controls (e.g., same chart language vs. different question/prompt language combinations). Although the discussion mentions two experiment settings, the main text omits key results and reproducibility details.
6. Figure 1’s caption inconsistently refers to “Task4&5,” while the text only defines four tasks; several English expressions and formatting issues need refinement. Such inconsistencies, though minor, weaken the authority expected from a benchmark paper.
7. Compared to existing multi-chart benchmarks, the proposed work still relies on a multiple-choice format. The inherent open-endedness of “strategy formulation/open QA” is discretized into fixed options, potentially reducing external validity for real applications. The paper does not convincingly demonstrate any “new diagnostic dimensions” for model capability or “direct contribution” to training/alignment research within the community.

**Questions:**

Questions are those listed under the Weaknesses section.

---

### Official Review · Reviewer_wQLy · 2025-10-28

**Soundness:** 3
**Presentation:** 3
**Contribution:** 2
**Rating:** 4
**Confidence:** 3

**Summary:**

This paper proposes a new benchmark MultiChartQA-R, which contains questions rangeing from fundamental abilities to decision-making capabilities. It also provides a new evaluation metric, $MF_{\beta}$, which is more flexible in application, followed by some observations out of experiments.

**Strengths:**

The overall writing logic of this article is very clear.
- It provides a novel benchmark which covers more complex questions compared to existing benchmarks.
- It proposes a new metric to evaluate the capability of anomaly and pattern analysis and strategy recommendation.
- This paper evaluates the performance of many MLLMs in multi-chart settings, and extracts some conclusions were summarized from the experiment.

**Weaknesses:**

There is a core concerns. Solving these concerns may influence my rating.
- The questions that evaluate the capability of anomaly and pattern analysis and strategy recommendation are actually turned into a true-or-false or selection question, where intuitively these should be a question of testing the model's generative capabilities. This change weakens the comprehensiveness of the model performance evaluation. Also this is away from Question Answering benchmark.

There are some other concerns but are not as important as concerns above:
- How to define easy and difficult items in interfering items.
- In the experiments, there are no methods that are designed for chart settings. This is a very important aspects to measure the validity of this benchmark.
- The first two tasks are manually annotated. This may lead to differences between the performance of the model and humans during the final evaluation. Why not just annotate this using other models, followed by human check?

**Questions:**

Please refer to section Weaknesses

---

### Official Review · Reviewer_XBqy · 2025-10-30

**Soundness:** 2
**Presentation:** 2
**Contribution:** 3
**Rating:** 4
**Confidence:** 4

**Summary:**

This paper introduces MultiChartQA-R, a multilingual benchmark for multi-chart question answering covering four progressively complex reasoning tasks: cross-chart trend inference, complementary data integration, anomaly and pattern analysis, and strategy recommendation. The benchmark includes 695 chart-code pairs and 2,160 QA pairs across English, Chinese, and Spanish, and also proposes a flexible multiple-choice evaluation metric to better assess multi-option reasoning behavior. The authors evaluate 13 proprietary and open-weight MLLMs and show substantial performance gaps compared to humans, particularly in cross-chart perception and data integration. The authors also provide extensive studies to test model robustness against different aspects.

**Strengths:**

- The methodology is generally sound. The construction pipeline combines chart-to-code reconstruction, controlled QA synthesis, multilingual chart/code translation with RAG assistance, and manual review for a subset of data. The metric is reasonable for multi-option multiple-choice settings. Evaluation covers a broad set of models. Claims about model limitations are supported by experiments.
- The discussion is particularly interesting, involving isolated experiments that evaluate models’ retrieval robustness when having irrelevant visual information and mixing questions and charts in a single conversation — both of which mimic real-world scenarios. The robustness on language mixing is also well-experimented that both multilingual charts and multilingual text scenarios are covered.

**Weaknesses:**

- The multilingual analysis is interesting, especially the distinction between varying languages in charts vs. varying languages in QAs/prompts, but the current discussion feels shallow. Instead of only stating which settings lead to larger fluctuations, I would like to see more rigorous analysis. Concretely, different models show different degrees of performance fluctuation, and some fluctuations appear small. Which differences are statistically significant? Also, some models seem less robust when chart text languages change, while others struggle more when QA language changes. It would be valuable to analyze and justify why this is the case.
- Section 4.3 and Appendix G are poorly written and difficult to follow. It would help to include a visual illustration or clearer explanation highlighting the differences among the settings. Providing error bars or variance estimates for accuracy curves would clarify statistical significance.
- The human evaluation process is vague. While the authors state that 30% of the data was inspected, it is unclear how human performance numbers were obtained. More detail on evaluator background, how answers were collected (e.g., did annotators see exactly the same inputs as models?), and the number of annotators per question would help assess reliability.
- While the benchmark aims to test reasoning-heavy abilities, models already appear to saturate on the latter two question types (relative to human performance), suggesting these tasks may not be sufficiently challenging.
- Chart type distribution appears imbalanced. Although 14 chart types are included, 9 chart types represent <5% of the dataset, while the remaining 5 chart types account for >95%, with bar charts (bar and stacked bar) alone exceeding half of the dataset. This limits representativeness. Also, single chart looks relatively simple based on the authors’ provided chart images in the supplementary material.

**Questions:**

- Figure 2 is visually complex and difficult to read (arrows, fonts, layout). Could the authors revise or simplify this figure?
- In Table 4, adding irrelevant charts hurts trend inference and data integration the most, but has limited effect on anomaly/pattern attribution and strategy recommendation. Could the authors explain why? If retrieval is the bottleneck for the first two tasks, does this imply the latter two may not strongly require chart grounding? I am curious about authors' thoughts.
- How do the authors ensure translation validity across languages? More detail on multilingual quality control would be helpful.

---

### Official Review · Reviewer_Apjk · 2025-11-03

**Soundness:** 3
**Presentation:** 3
**Contribution:** 2
**Rating:** 2
**Confidence:** 4

**Summary:**

This paper introduces MultiChartQA-R, a new benchmark dataset for evaluating the multi-chart question-answering capabilities of Multimodal Large Language Models (MLLMs). The proposed benchmark is available in three languages, consisting of 2,160 QA pairs across 695 charts, structured into four reasoning tasks of increasing complexity: cross-chart trend comparison, complementary data integration, anomaly and causal analysis, and strategy recommendation. The authors also proposed a flexible multiple-choice evaluation metric. The authors evaluate 13 MLLMs and find significant performance gaps compared to humans in this benchmark.

**Strengths:**

1. The paper is well motivated. It addresses the current limitations on chart-related benchmarks, which are mostly limited to single charts. To address this, the authors have proposed a multi-chart benchmark.

2. The four-task structure is also well-motivated, targeting real-world use cases.

3. Multilingual extension.

4. The evaluation of 13 models provides a strong baseline for the community.

5. Proposed a flexible evaluation metric.

**Weaknesses:**

1. The paper claims to use charts from real-world reports. However, the authors used the chart-to-code technique, where an LLM converts each chart into Python code. While this is an innovative solution, the benchmark does not test reasoning on real-world data. Rather, it seems more of an LLM-generated synthetic benchmark that mimics the appearance of real-world charts.

2. Moreover, synthetic data generation was also followed in complex tasks (Task 3 and Task 4) by using a reasoning model. But no information about this model was provided.

3. Missing details regarding the RAG web sources and the model being used.

4. The evaluation pipeline for Task 2 uses DeepSeek-V3.1. Why this model is used? What is the limitation of this model? Why Regex not used? Human evaluation was also needed in this step to justify the selection of DeepSeek.

5. Justify the novelty of the proposed benchmark in comparison this: https://arxiv.org/abs/2410.14179

6. Lack of justification of the effectiveness of the proposed evaluation metric in comparison to other metrics. Compare its effectiveness over Precision, Recall, F1 etc.

**Questions:**

Check the weaknesses. Address them.

---

### Note · Authors · 2025-11-16

**Comment:**

**Title: Withdrawal of Submission**

Dear Area Chair and Reviewers,

Thank you very much for your time and effort in reviewing our manuscript. We have carefully read all the reviews. After thorough consideration, we have decided to withdraw our paper from ICLR 2026.

The reviewers' comments are very insightful and have helped us identify several areas for significant improvement. We plan to incorporate this valuable feedback to substantially revise our work.

We sincerely appreciate the opportunity to submit to ICLR 2026 and are grateful for the high-quality feedback provided by the reviewers.

Best regards,

The Authors

**Withdrawal Confirmation:**

I have read and agree with the venue's withdrawal policy on behalf of myself and my co-authors.